# Allelic variation of Avr genes in highly virulent strains explains severe wheat stem rust epidemics

Rebecca E. Spanner [1,6], Eva C. Henningsen [2,6], Camilla Langlands-Perry [2,6], Jian Chen [2], Jibril Lubega[3], Oadi Matny [1], David Lewis[2], Li Chen Cheah[2], Zhouyang Su[2], Alexis Feist[1], Eric S. Nazareno [1], Feng Li [1], Megan A. Outram [2], Taj Arndell [2], Thomas Vanhercke[2], Nino Virzi[4], Ming Luo [2], Michael Ayliffe[2], Eric Stone [5], Kostya Kanyuka [3], Jana Sperschneider [2], Peter N. Dodds [2] ✉, Brian J. Steffenson [1] ✉ & Melania Figueroa [2] ✉

Wheat stem rust is a disease of global importance caused by the fungal pathogen *Puccinia graminis* f. sp. *tritici* (*Pgt*). Here we generate chromosome-level, nuclear-phased genome references for *Pgt* isolates ETH2013-1 and ITA2018-1, representing races TKTTF and TTRTF respectively, that have caused major epidemics in Africa and Europe. The nuclear haplotypes of ETH2013-1 and ITA2018-1 are unique and unrelated to those of Ug99 and Pgt21. *Pgt* nuclear haplotypes show extensive variation in sequence and copy number of six known *Avr* genes and *AvrSr33*, which we identify through an effector gene library screen. Recognition properties of 22 novel *Avr* gene variants explain the race virulence phenotypes and the outbreak of TTRTF on durum cultivars containing *Sr13b*, since ITA2018-1 carries a homozygous deletion of *AvrSr13*. This work establishes an *Avr* gene atlas for *Pgt* that can inform wheat breeding and enable development of sequence-based virulence diagnostic tools for pathogen surveillance.

The sudden emergence of novel strains of plant pathogens that overcome previously resistant cultivars is a major challenge in the management of crop diseases[1]. In the case of wheat stem rust disease caused by *Puccinia graminis* f. sp. *tritici* (*Pgt*), the emergence of novel races has led to multiple epidemics in recent years resulting in substantial yield losses in the most vulnerable wheat-growing regions[2–4]. For instance, *Pgt* race TTKSK (known as Ug99) was first detected in Uganda in 1998[5] and caused major epidemics in eastern Africa during the 2000's. TTKSK and other races in the Ug99 lineage remain a significant threat for wheat production as they continue to spread into South Asia[6,7]. More recent epidemics in Africa and Europe have been caused by other races of *Pgt* with different virulence profiles, including TKTTF and TTRTF[8,9], presenting further challenges for disease resistance breeding.

Genetic resistance to the rust fungi in small grain cereal hosts is typically conferred by race-specific resistance (*R*) genes encoding immune receptors that recognise pathogen effector proteins encoded by avirulence (*Avr*) genes in a gene-for-gene manner[10,11]. Thus, the evolution of new rust races relies on genetic changes such as mutation, somatic nuclear exchange and sexual recombination[12,13]. Over 20 stem rust resistance genes have been isolated from wheat and barley[14] and the identification of *Avr* genes from *Pgt* is accelerating with the

---

[1]Department of Plant Pathology, University of Minnesota (UMN), St. Paul, MN, USA. [2]Commonwealth Scientific and Industrial Research Organisation (CSIRO), Canberra, ACT, Australia. [3]Niab, Park Farm, Villa Rd, Impington, Histon, Cambridge, UK. [4]Council for Agricultural Research and Economics (CREA), Research Centre for Cereal and Industrial Crops, Acireale, CT, Italy. [5]Research School of Biology, The Australian National University (ANU), Canberra, ACT, Australia. [6]These authors contributed equally: Rebecca E. Spanner, Eva C. Henningsen, Camilla Langlands-Perry. ✉e-mail: peter.dodds@csiro.au; bsteffen@umn.edu; melania.figueroa@csiro.au

emergence of new tools and improved genome resources[15,16]. Currently six *Avr* genes have been identified: *AvrSr13*, *AvrSr22*[15], *AvrSr27*[17], *AvrSr35*[18], *AvrSr50*[19], and *AvrSr62*[20]. However, there is a need to systematically characterise the diversity and function of allelic and paralogous variants of these *Avr* genes in *Pgt* populations, particularly in major epidemic-causing lineages. For example, Ortiz et al.[21] examined *AvrSr50* allelic diversity by assembling sequence variants from Illumina sequence data for a range of isolates, identifying 14 different sequence variants. However, the dikaryotic nature of rust fungi, with each isolate containing two independent haploid nuclei, has been a challenge for broader *Avr* genotype assignment[13]. Recently, the combination of Hi-C chromatin contact analysis and highly accurate long-read DNA sequencing has enabled resolution of the two haploid genomes and assembly of nuclear-phased genome references for several rust species[22–26]. This approach was first adopted for *Pgt*, generating nuclear haplotype-phased assemblies for isolates Ug99 and Pgt21-O[22], which belong to clonal lineages designated as clades I and IX, respectively, based on SNPchip and sequence-based phylogenies of global isolates[27]. Genome comparison revealed that these two isolates share one near-identical nuclear haplotype genome (designated haplotype A), while the second nuclear haplotype in each isolate (C and B, respectively) are highly divergent[22]. These data suggest that the Ug99 (clade I) lineage (nuclear haplotypes A and C) arose via somatic hybridization and nuclear exchange between a clade IX isolate (nuclear haplotypes A and B) and an unknown donor containing haplotype C. Notably, an isolate collected in Iran in 2015 (IR-01) was shown to contain the C genome and could represent the other parental lineage of Ug99[22]. Guo et al.[27] also identified a lineage of *Pgt* (designated as clade II) present in Africa prior to the emergence of Ug99 as a candidate for the C genome donor based on a genome admixture analysis. However, the relationship between clade II and IR-01 is not known.

Other important new races of *Pgt* have emerged more recently. In 2013, an epidemic broke out in Ethiopia caused by race TKTTF that possesses novel virulence for the resistance gene *SrTmp* present in the widely grown wheat cultivar Digalu[8]. Based on genotyping with a SNPchip (~1000 nucleotide polymorphisms), these isolates of race TKTTF were divided into two genetic groups, designated as clades IV-A and IV-B, raising questions about their relationship[8]. Interestingly, clade IV-A isolates of race TKTTF had also been detected in Turkey in 2012[28], Germany in 2013 and Georgia in 2014[2], suggesting an earlier origin and migration to Ethiopia of this lineage. Patpour et al.[29] detected TKTTF isolates of clade IV-B in Europe from 2017 onwards. Lewis et al.[30] also described an isolate of race TKTTF from the UK in 2013. Subsequent analysis of isolates collected during stem rust outbreaks in Germany and Georgia led to the designation of six additional clade IV subgroups[2,31].

In 2016, a wheat stem rust outbreak occurred in Sicily caused by a novel *Pgt* race, TTRTF[9], which overcame the *Sr13b* resistance in both durum and bread wheat[2,32]. Race TTRTF subsequently became the most prevalent *Pgt* virulence type in Europe between 2017 and 2021[29] and has also been detected in Iran[33] and Ethiopia[34]. Olivera et al.[2] also detected isolates of race TTRTF in Georgia in 2014 and 2015, again suggesting a broader dispersal prior to the epidemic in Sicily. The TTRTF isolates were assigned to clade III-B by SNPchip and simple-sequence repeat marker analysis and are genetically distinct from the TKTTF clade IV isolates[2,29].

To investigate the genomes and virulence genotypes of *Pgt* races causing these recent epidemics, we generated haplotype-phased genome assemblies for isolates from Ethiopia (ETH2013-1 of race TKTTF clade IV-B subgroup) and Sicily (ITA2018-1 of race TTRTF). These genome resources enabled us to identify variants of known *Avr* genes in these isolates, assign their recognition properties through functional assays, and corelate their genotypes to virulence phenotypes. To facilitate sequence-based detection of virulence genotypes in *Pgt* lineages and enhance surveillance capabilities, we initiated an *Avr* gene atlas that includes *Avr* gene variants and associated recognition profiles.

## Results

### Haplotype-phased genome references for two *Pgt* isolates responsible for recent epidemics

Two *Pgt* isolates were obtained from regions where recent epidemics occurred: ETH2013-1 was collected during the 2013 epidemic growing on wheat cultivar 'Digalu' in Ethiopia and ITA2018-1 was collected in 2018 in Sicily, Italy. Inoculation of these isolates onto a standard wheat differential set confirmed their races as TKTTF and TTRTF, respectively (Supplementary Data 1). Both isolates were also inoculated onto transgenic wheat lines containing the stem rust resistance genes *Sr13c*, *Sr22*, *Sr27*, *Sr33*, *Sr35*, or *Sr50* genes and the wheat introgression line Zahir 1644, which carries *Sr62*. ITA2018-1 was virulent on *Sr13c* and *Sr35*, but avirulent on the other *Sr* genes, while ETH2013-1 was avirulent on all lines (Supplementary Fig. 1). To determine genetic relationships of these isolates to those previously characterized, whole genome short read sequence data were generated and used in a phylogenetic analysis with publicly available sequences for 144 *Pgt* isolates, including 105 with whole genome data, 38 with RNAseq data and one with both whole genome and RNAseq data (Fig. 1, Supplementary Fig. 2). ETH2013-1 was placed in a clonal lineage containing multiple European and African isolates[30,35], including 13ETH23-1, which was one of the originally defined isolates of race TKTTF identified in Ethiopia and assigned to clade IV-B based on a SNPchip assay[8]. This clade also includes the isolate UK-01 (race TKTTF) detected in the UK in 2013[30]. Other isolates collected from Ethiopia, also with race TKTTF, fall into a separate clonal lineage that is genetically divergent from the clade IV-B and includes isolates previously assigned to clade IV-A by SNPchip analysis[8]. The IV-A clade also includes an isolate from Israel collected in 2012 (12ISR2083), further confirming the wider dispersal of this lineage prior to the Ethiopian epidemic. ITA2018-1 groups in a clonal lineage with four isolates (IT-01, IT-02, IT-03, IT-04) collected in Sicily during the 2016 epidemic but with unknown race types[30].

PacBio HiFi and Hi-C sequencing data were generated for ETH2013-1 and ITA2018-1 and used to assemble nuclear haplotype-phased reference genomes for each isolate using Hifiasm (Supplementary Data 2). This resulted in two distinct haplotypes for each isolate, with no haplotype phase switches detected by NuclearPhaser[23] in contigs of either assembly. Contigs were scaffolded using Hi-C contact data resulting in 18 chromosomes in each haplotype, with between 4 and 14 gaps per haplotype (Supplementary Data 3). Individual chromosomes in the ETH2013-1 and ITA2018-1 assemblies showed an average of 97.8% of *trans* Hi-C contacts to other chromosomes in the same designated nuclear haplotype (Supplementary Data 4; Supplementary Fig. 3), confirming accurate phasing and nuclear assignment. This was also supported by whole-genome Hi-C contact maps (Supplementary Fig. 4). Unplaced contigs were mostly small (N50 of 35.6 kbp and 39.7 kbp for ETH2013-1 and ITA2018-1, respectively) and had extremely high repeat content (82.33% and 89.29%; Supplementary Data 3). Nuclear haplotype sizes ranged from 82.9 to 86.5 Mbp (average 84.8 Mbp), similar to the previously assembled haplotypes for Pgt21-O (83.9 and 85.9 Mbp)[22] (Supplementary Data 3).

Here, we switched to a haplotype numbering scheme rather than maintain the A, B and C haplotype nomenclature established by Li et al.[22] for isolates Pgt21 and Ug99. This allows for the future expansion of the *Pgt* pangenome to more than 26 haplotypes, noting that for instance over 50 haplotypes have now been assembled for the related species *Puccinia coronata* f. sp. *avenae* (*Pca*)[24,36,37]. We designated the A, B and C haplotypes of Pgt21 and Ug99 as hap01, hap02 and hap03 respectively, while the two haplotypes of ETH2013-1 were designated as hap04 and hap05 and those of ITA2018-1 as hap06 and hap07 (Fig. 2A). We also obtained Hi-C data for the *Pgt* isolate Ug99 and used this to scaffold the published contigs[22] into chromosome assemblies for its hap01 and hap03 nuclear haplotypes (Supplementary Data 5). Genome alignments showed that the chromosomes for each haplotype were similar in size and highly syntenic in structure with no

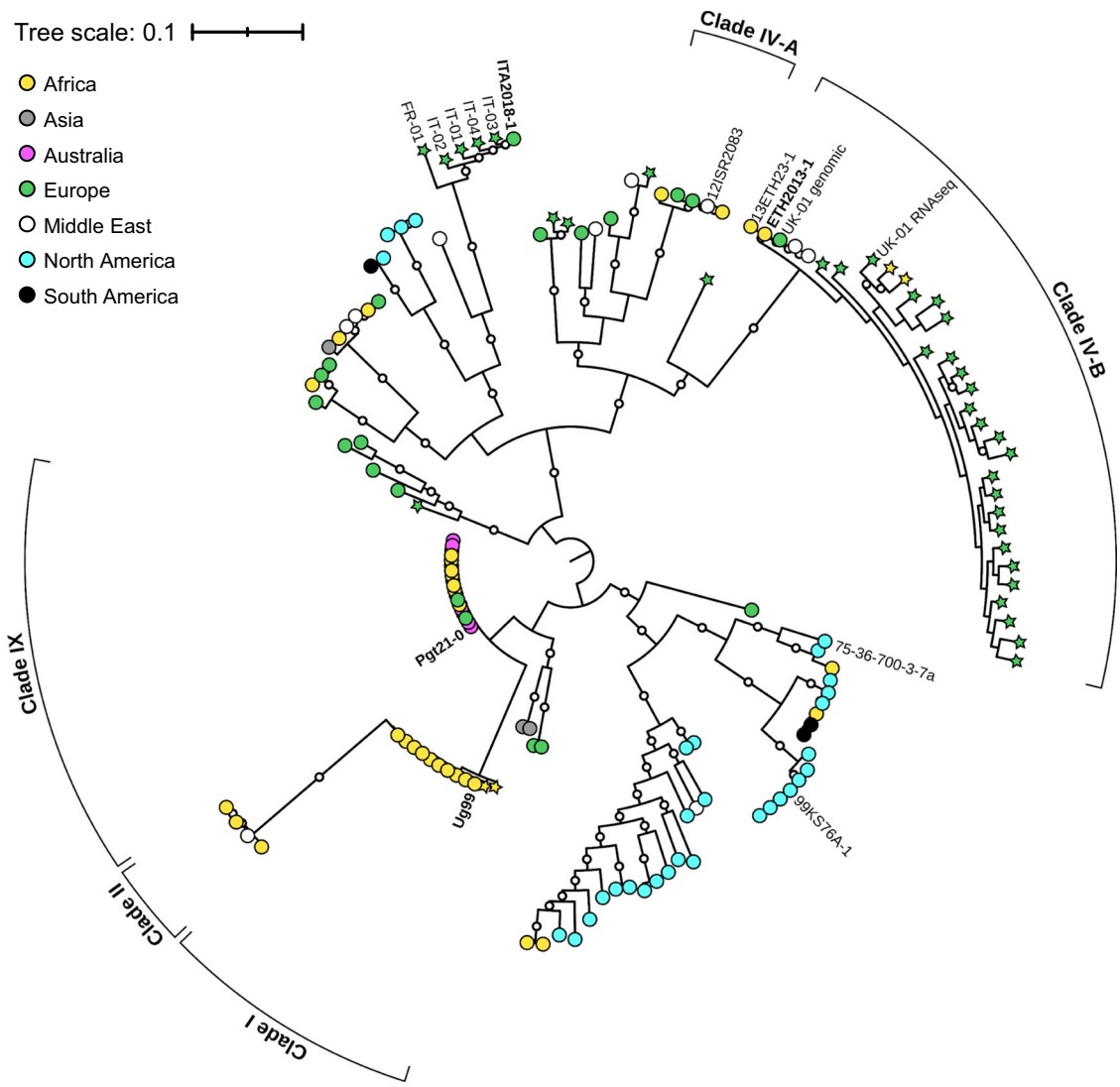

**Fig. 1 | Genotypic relationships among *Puccinia graminis* f. sp. *tritici* (*Pgt*) isolates from diverse geographic locations.** Maximum likelihood phylogenetic tree of 146 *Pgt* isolates based on 393,568 SNPs compared to the diploid Pgt21-0 reference assembly. Nodes are colored by the geographic region where isolates were collected. Node symbol indicates sequence data type: circle = whole genome sequencing, star = RNA-seq. Bootstrap values over 80% are shown as circles at branch midpoints. Scale is mean substitutions per site. Nuclear-phased references exist for isolates with bold names. Clade identifiers are based on previous reports using SNPchip analysis[2,8,31].

major rearrangements (Fig. 2B, Supplementary Fig. 5), except for hap02 which contains two large translocations between chromosomes 3 and 5 and between 8 and 16 as previously reported[22]. The BUSCO completeness scores of 95.6–96.4% and 0.8–1.1% duplication for each haplotype (Supplementary Data 3) were comparable to the haplotype-phased genomes obtained previously for *Pgt*[22], *P. triticina*[23,38], *P. striiformis* f. sp. *tritici*[39] and *Pca*[24,36,37]. Pairwise whole genome alignment of haplotypes showed divergence between all pairs, with haplotype identity lowest between hap02 and hap07 (88.46% bases aligned, 95.09% identity), while hap02 and hap03 were the most similar (92.12% bases aligned, 97.63% identity) (Supplementary Data 6). Recombination block analysis revealed no evidence for recent recombination events, except for a few small regions ($n = 22$, 8.5 Mbp total length) shared between hap05 and hap07 (Supplementary Fig. 6; Supplementary Data 7).

### Variation at mating type loci in the newly designated haplotypes of *Puccinia graminis* f. sp. *tritici*

Mating type alleles at both the *a* and *b* loci were identified in all newly designated haplotypes (Supplementary Fig. 7A, B; Supplementary Data 8) using homology searches of previously published *Pgt* sequences[22,40]. Haplotypes hap05 and hap07 encode the *STE3.2.2 a* locus allele with the associated *mfa2* pheromone precursor located 845 and 878 bp downstream, respectively. Hap04 and hap06 encode the *STE3.2.3* receptor allele with the *mfa3* pheromone precursor located 31,037 and 29,864 bp upstream, respectively. The *STE3.2.2* and *mfa2* allele in each isolate is identical to the sequence published for *Pgt* isolate 75-36-700-3-7a[40]. However, the *STE3.2.3* alleles from isolates Pgt21-0, ETH2013-1, and ITA2018-1 differ from 75-36-700-3-7a by ten nucleotides including four non-synonymous substitutions leading to amino acid differences (I332M, H365Q, D373N, and Q376P). The Ug99 *STE3.2.3* allele encodes a protein truncated by 48 amino acids due to a single nucleotide deletion as reported by Li et al.[22]. Several clade I isolates share this mutation (04KEN15604, 06KEN19V3, 07KEN24-4, 14KEN58-1, Pgt-60, ET-01, and ET-02), although some others (Pgt-55, Pgt-59, Pgt-61, 72ETH11-4; Supplementary Data 9) do not, suggesting a recent mutation event in this lineage (Supplementary Fig. 7C). Interestingly, this polymorphism corresponds to the separation between the *Sr31* avirulent and *Sr31* virulent isolates in this lineage. The *mfa3* alleles all share the same single amino acid difference (N27D) from the

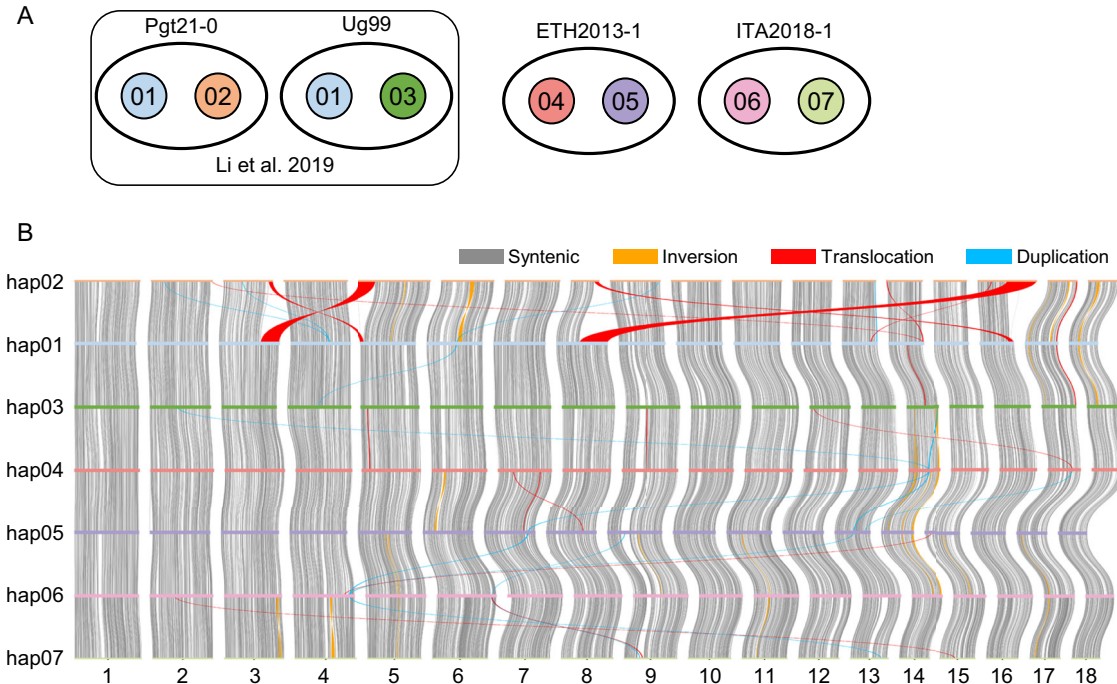

**Fig. 2 | Sequence collinearity among chromosomes of *Puccinia graminis* f. sp. *tritici* (*Pgt*) nuclear haplotypes. A** Haplotype designations (01 to 07) for haploid nuclei of the four haplotype-phased *Pgt* isolates. **B** Synteny plot of seven haplotypes (hap01 to hap07), with chromosomes represented on the *x*-axis. Hap02 is shown first as it has two large chromosomal translocations relative to the other haplotypes. Vertical coloured lines connecting chromosomes reflect four relationships between sequences: syntenic (grey), inverted (yellow), translocated (red), and duplicated (blue).

published 75-36-700-3-7a allele[40]. *K*-mer containment for both of the *a* mating locus alleles (*STE3.2.2* and *STE3.2.3*) was high in genomic and RNAseq sequence data from almost all isolates from the global collection, indicating uniform heterozygosity for these two alleles (Supplementary Data 9). Two *Pgt* isolates showed low shared *k*-mers for both *STE3.2* alleles (69.10–87.60%), but this was likely due to their low mapped read coverage (76WA1295A – 3.5X, DE-01 – 6.38X). The *k*-mer identity for *STE3.2.3* was high in all samples, but percentage of shared *k*-mers was lower (73.20-89.90%) in 63 isolates. Comparisons of read mapping revealed conserved patterns of either nine or ten SNPs within *STE3.2.3* exons in these isolates. The gene sequences with 10 SNPs matched the published *STE3.2.3* allele from 75-36-700-3-7a, while those with nine differed by a single SNP. Both variants of *STE3.2.3* are present in isolates from Asia, Africa, and Europe, while to date all isolates from North and South America have the 75-36-700-3-7a variant, and isolates from Australia possess only the hap01 variant.

The four haplotypes from ETH2013-1 and ITA2018-1 each have unique and novel *b* locus allele pairs (*b5* to *b8*), bringing the total of known *b* locus alleles represented in *Pgt* genome references to eight (Supplementary Fig. 7A, B)[22,27,40]. Using a *k*-mer containment approach, one or two known *b* locus alleles were assigned to 113 of the 146 *Pgt* isolates, with all isolates showing heterozygosity at this locus (Supplementary Fig. 2; Supplementary Data 10). These data support the hypothesis of mating type being controlled by a bi-allelic *a* locus and multi-allelic *b*-locus as observed in other species of *Puccinia*[36,38].

### Identification of *Puccinia graminis* f. sp. *tritici* lineages with shared nuclear haplotypes

To identify potential somatic hybridization events between *Pgt* isolates/lineages, we performed *k*-mer containment analysis[36,38] on all the available whole genome sequence data. Isolates in the Ug99 and Pgt21-0 lineages fully contain the *k*-mers from hap01, while two isolates from Pakistan and two from the Czech Republic fully contain *k*-mers from hap02 (Fig. 3; Supplementary Data 11), consistent with the haplotype specific phylogeny results reported previously[22]. All clade II isolates show full containment of hap03 *k*-mers. This includes isolate IR-01 (collected in Iran in 2015), which Li et al.[22] identified as a carrier of hap03, as well as isolates from east Africa that predate Ug99 emergence and were postulated by Guo et al.[27] as a possible donor lineage of hap03 to Ug99. These shared haplotype assignments were also supported by haplotype-specific phylogenetic trees (Supplementary Fig. 8A–C).

The *k*-mer containment analysis also showed that hap04 and hap05 (ETH2013-1) were shared by all the isolates placed in the clade IV-B clonal lineage (Fig. 1; Fig. 3; Supplementary Data 11). Hap05 was also shared by the Iranian isolate IR-06 (100% *k*-mer identity, 99.92% shared *k*-mers), which agrees with the phylogenetic tree based on the hap05 reference (Supplementary Fig. 9A, B; Supplementary Data 11). No isolates were identified that shared hap06 or hap07 of ITA2018-1 by *k*-mer containment analysis (Supplementary Data 11) or using haplotype-specific trees (Supplementary Fig. 9C, D). Results from the *b* mating locus allele assignments were consistent with the postulated presence of hap02 in Czech and Pakistani isolates and hap05 in IR-06, as these isolates contained the *bW1bE1* (hap02) or *bW6bE6* (hap05) alleles respectively (Supplementary Figs. 8B, 9B; Supplementary Data 10).

### Variation in avirulence effector genes underlies virulence phenotypes

To understand the genotypic differences behind the virulence phenotypes of isolates ETH2013-1 and ITA2018-1, and build an *Avr* gene catalogue, we examined sequence variation in known *Avr* genes and adopted a consistent nomenclature to refer to these variants. *Avr* variants in the reference genomes were assigned a two-digit identifier based on their unique protein sequence, with previously characterized variants renamed to fit this scheme (e.g., AvrSr50-A1 became AvrSr50-01) (Supplementary Data 12). Variants encoding the same protein sequence but with synonymous nucleotide changes were

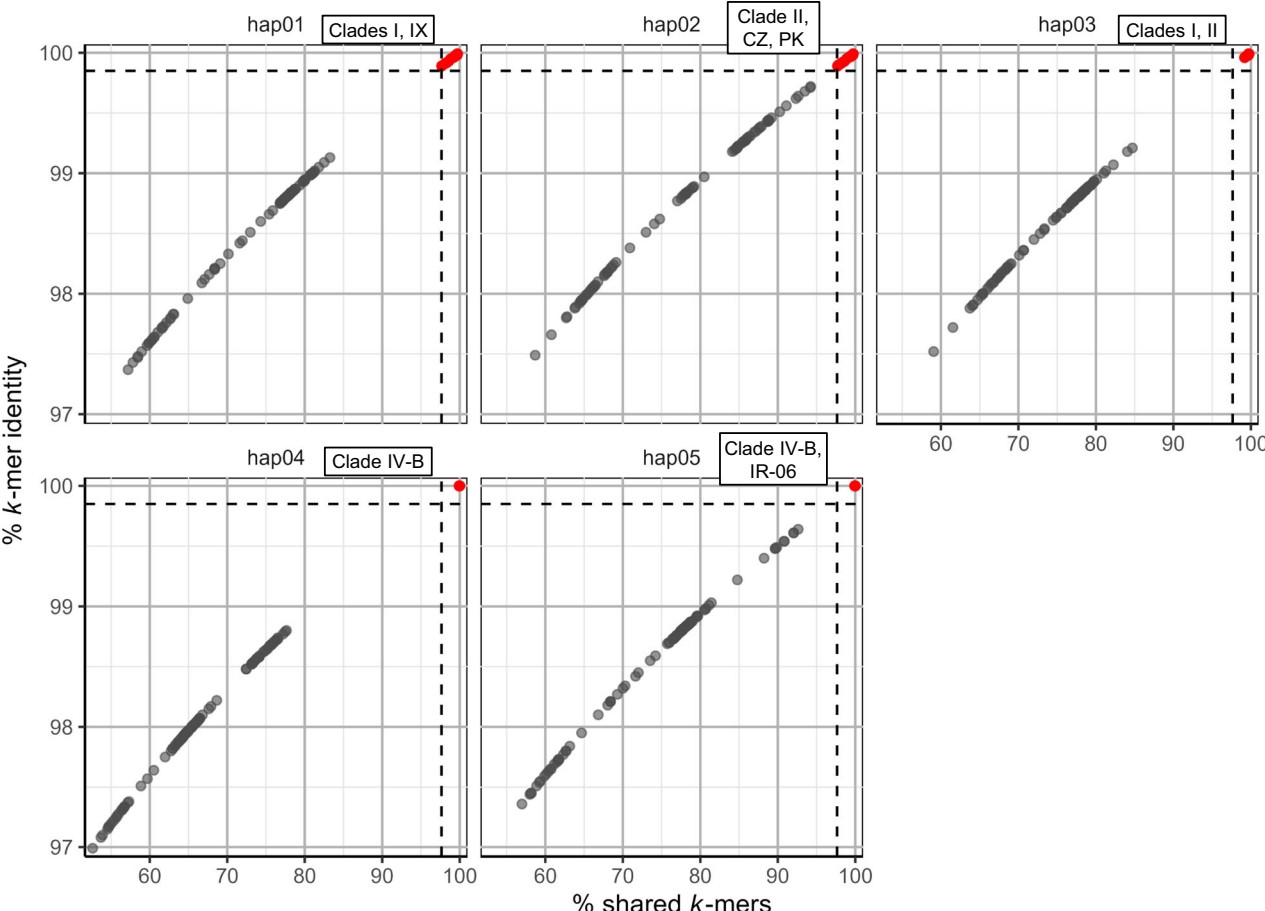

**Fig. 3 | *K*-mer containment of phased nuclear haplotypes in global *Puccinia graminis* f. sp. *tritici* isolates with available short-read sequencing data.** The containment of hap01 and hap02 (Pgt21-0), hap03 (Ug99), and hap04 and hap05 (ETH2013-1) in whole genome short-read sequences from 108 *Pgt* isolates. The *k*-mer identity (%) score was plotted against the shared *k*-mer (%) score for containment of the phased haplotypes for each isolate. Coloured data points represent isolates containing the respective haplotype (*k*-mer identity ≥ 99.89% and shared *k*-mers ≥ 97.67%), and data points labelled in grey represent isolates that do not contain that haplotype. Raw *k*-mer containment values are provided in Supplementary Data 11.

distinguished by an additional suffix (e.g., *AvrSr22-02.1* and *AvrSr22-02.2*). Figure 4 shows a summary of the nuclear-resolved genotypes of isolates Pgt21-0, Ug99, ETH2013-1 and ITA2018-1 for known *Avr* loci in *Pgt*. Detailed maps of *Avr* loci are shown in Supplementary Figs. 10–14.

At the *AvrSr13* locus (Supplementary Fig. 10), ETH2013-1 has two novel gene variants that encode proteins (AvrSr13-02, -03) that differ by a single amino acid in the signal peptide region and both differ by six amino acids from AvrSr13-01[15]. However, ITA2018-1 has null alleles for this locus in both haplotypes due to deletion events. This explains the virulence of this isolate/lineage on *Sr13*[2,32] (Supplementary Data 1). For *AvrSr22*, both ITA2018-1 and ETH2013-1 isolates carry one allele encoding an identical protein sequence to AvrSr22-02[15] and one unique allele encoding a protein (AvrSr22-03) with a single amino acid difference from AvrSr22-02 (Supplementary Fig. 11). Substantial variation exists at the *AvrSr27* locus, with differences in the encoded amino acid sequence (six novel alleles detected in ETH2013-1 and ITA2018-1), gene copy number (one or two per haplotype) and gene orientation (Fig. 4; Supplementary Fig. 12). The *AvrSr50/AvrSr35* locus in all haplotypes from ETH2013-1 and ITA2018-1 is most structurally similar to hap02 (Supplementary Fig. 13). ETH2013-1 is homozygous for *AvrSr50-03* (previously *AvrSr50-A2*) which is recognised by *Sr50*[21] and explains its avirulent phenotype on this resistance gene. ITA2018-1 is heterozygous for a novel recognized allele *AvrSr50-04* (see below) and the known recognized *AvrSr50-05* allele (previously AvrSr50-B4). ETH2013-1 is homozygous for *AvrSr35-01*, which is identical to the originally isolated allele from 99KS76[18], while ITA2018-1 is

homozygous for the novel *AvrSr35-02* allele, with unknown recognition. The *AvrSr62* locus in Pgt21-0 is complex, with four paralogs in hap01 and three in hap02, of which two in each haplotype are recognized (01, 04, 05 and 07)[20]. Hap03, hap04, hap06 and hap07 contain similar *AvrSr62* haplotypes to hap02, each containing three paralogs with some minor sequence differences (Supplementary Fig. 14). The *AvrSr62* locus in hap05 is similar to that in hap01 but with one additional intact gene copy (*AvrSr62-09)* that corresponds to a pseudogene in hap01 (inactive due to a splice site mutation). Altogether, four novel variants of *AvrSr62* (-08 to -11) were identified.

To assess the potential avirulence function of the effector proteins encoded by the novel *Avr* gene variants, we generated expression constructs for each variant and tested them by co-expression with the corresponding *Sr* genes in wheat protoplasts and in *Nicotiana benthamiana*. Proteins AvrSr13-02, AvrSr22-03, AvrSr27-05 to -10, and AvrSr50-04 all triggered cell death when co-expressed with the corresponding Sr protein but not when expressed alone, confirming their recognition (Fig. 5; Supplementary Figs. 15–19). Likewise, recombinant Barley Stripe Mosaic Virus (BSMV) expressing any of these *Avr* gene variants was unable to infect wheat lines containing the corresponding *Sr* genes (Fig. 5B; Supplementary Figs. 15, 17, 19). The AvrSr35-02 protein failed to trigger cell death when co-expressed with *Sr35* in *N. benthamiana* and induced a weaker response in protoplasts than AvrSr35-01, consistent with the *Sr35*-virulent phenotype of ITA2018-1 (homozygous for *avrsr35-02*) (Supplementary Fig. 20A–C). There are 40 single amino acid differences and 2 indels between AvrSr35-01 and

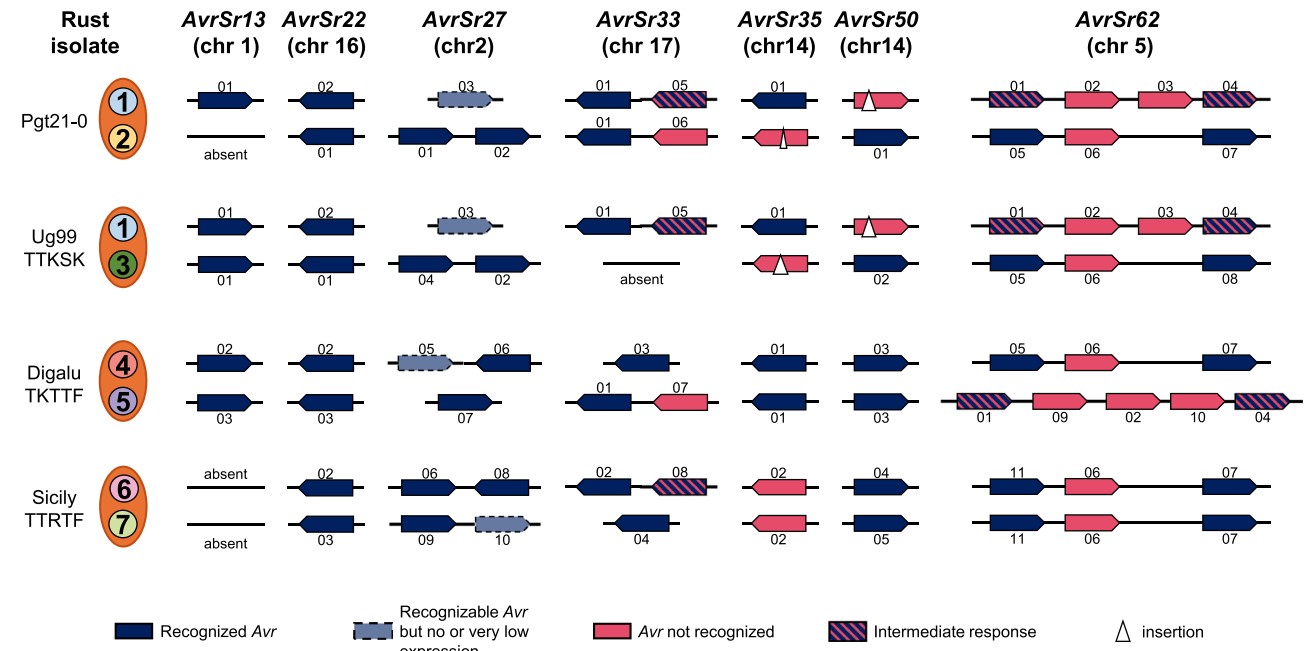

**Fig. 4 | Variation in *Avr* gene haplotypes in nuclear-phased *Puccinia graminis* f. sp. *tritici* (*Pgt*) genomes.** Schematic showing *Avr* gene variants encoded by haplotypes of the *AvrSr13*, *AvrSr22*, *AvrSr27*, *AvrSr33*, *AvrSr35/50*, *AvrSr62* loci in *Pgt* isolates Pgt21, Ug99, ETH2013-1 and ITA2018-1. Nuclear haplotypes of each strain are shown at left. Genes are shown as arrows and numbers indicate variant designation. Blue denotes *Avr* variants recognized by the corresponding *Sr* gene in functional assays, while pink denotes unrecognized variants. Blue and pink striped arrows denote *Avr* variants which gave an intermediate response in functional assays. Light blue fill with dashed outline indicates recognized alleles with zero or very low expression. Triangles represent insertional mutants previously described in ref. 22. Gene IDs for *Avr* variants are provided in Supplementary Data 12.

-02, but only one of these lies in the interaction interface with Sr35[41,42], S349 (AvrSr35-01) to R352 (AvrSr35-02). Experimental data in *N. benthamiana* shows exchanging this single residue is sufficient to abolish recognition in AvrSr35-01 and restore recognition in AvrSr35-02 (Supplementary Fig. 21D, E). For the *AvrSr62* locus, each haplotype contained two variants that were both recognised when co-expressed with *Sr62* in *N. benthamiana* or wheat protoplasts (Supplementary Fig. 21), and one to three unrecognized variants, similar to Pgt21-0.

Given that virulence on *Sr27* was associated with low expression of the *AvrSr27-3* allele despite its recognition in functional assays[17], we examined the expression levels for the corresponding variants in the two isolates during host infection. Both ETH2013-1 and ITA2018-1 possess at least one gene copy that was expressed, with most copies showing TPM > 20. There were two exceptions; *AvrSr27-05* (hap04) was not expressed (TPM < 1), and *AvrSr27-10* (hap07) had very low expression (average TPM = 5.21) during infection. Both gene variants occur in haplotypes with an adjacent expressed gene; thus, it is likely that both strains are homozygous for avirulence at this locus.

Thus, ETH2013-1 contains avirulence alleles corresponding to each of these *Sr* genes, consistent with its avirulent infection phenotype (Supplementary Data 1), while ITA2018-1 contains avirulence alleles for *Sr22*, *Sr27*, *Sr50* and *Sr62*, but is homozygous for virulence alleles for *Sr13* and *Sr35*, consistent with its infection profile.

### Identification of *AvrSr33* and characterization of genetic variants

To further expand the *Avr* gene catalogue, we also used the *Sr33* resistance gene[43] to screen a library of 1,373 effector gene candidates from Pgt21-0 by co-expression in wheat protoplasts[15,20]. A single effector gene construct (clone #0956) showed significantly reduced expression in the presence of *Sr33* relative to an empty vector control (Fig. 6A; Supplementary Fig. 22), suggesting immune recognition-induced protoplast cell death. We further tested this candidate (AvrSr33-01) by individual co-expression with *Sr33* in wheat

protoplasts, in *N. tabacum* and in *N. benthamiana*, as well as by VOX in wheat (Fig. 6B–E), which confirmed its specific recognition. The *AvrSr33* locus contains 0, 1, or 2 homologous copies of this gene in each haplotype, with a total of eight sequence variants identified (Fig. 4, Supplementary Fig. 23). Of these, *AvrSr33-02*, *-03* and *-04* were each recognized strongly by *Sr33* in co-expression assays in *N. benthamiana* and wheat protoplasts, *AvrSr33-05* and *-08* led to weak responses both in protoplasts and *N. benthamiana*, while *AvrSr33-06* and *-07* were not recognized in either assay (Fig. 6; Supplementary Figs. 24, 25). Thus, isolates Pgt21-0, ETH2013-01 and ITA2018-1 are assigned as homozygous for avirulence to *Sr33*, while Ug99 is heterozygous with a null allele in hap03 due to a 3,109 bp deletion relative to hap06.

## Discussion

Nuclear haplotype-phased reference genomes in cereal rust fungi are crucial for understanding genetic diversity in these dikaryotic organisms[44]. Previously, only two nuclear haplotype phased references were available for *Pgt*, derived from isolates Pgt21-0 and Ug99, with only the former scaffolded to chromosome level[22]. Here, we have expanded the genomic resources for *Pgt* by generating haplotype-resolved reference genomes for two isolates (ETH2013-1 and ITA2018-1) representing lineages that caused recent epidemics in Ethiopia and Italy[8,9] and which have both spread to become predominant races in Europe and Africa[29]. We also assembled full chromosomes for the Ug99 haplotypes. These resources have enabled the establishment of an *Avr* gene atlas compiling genotypic variation in known *Avr* loci across these important *Pgt* strains.

The four nuclear haplotype sequences of ETH2013-1 and ITA2018-1 are unique and highly diverged from those of the previous reference genomes (Ug99 and Pgt21-0). Thus, there are now a total of seven fully phased nuclear haplotypes available for *Pgt* (designated hap01 to hap07), with hap01 shared between Pgt21-0 and Ug99 due to a nuclear exchange event[22]. Another chromosome-scale genome assembly was

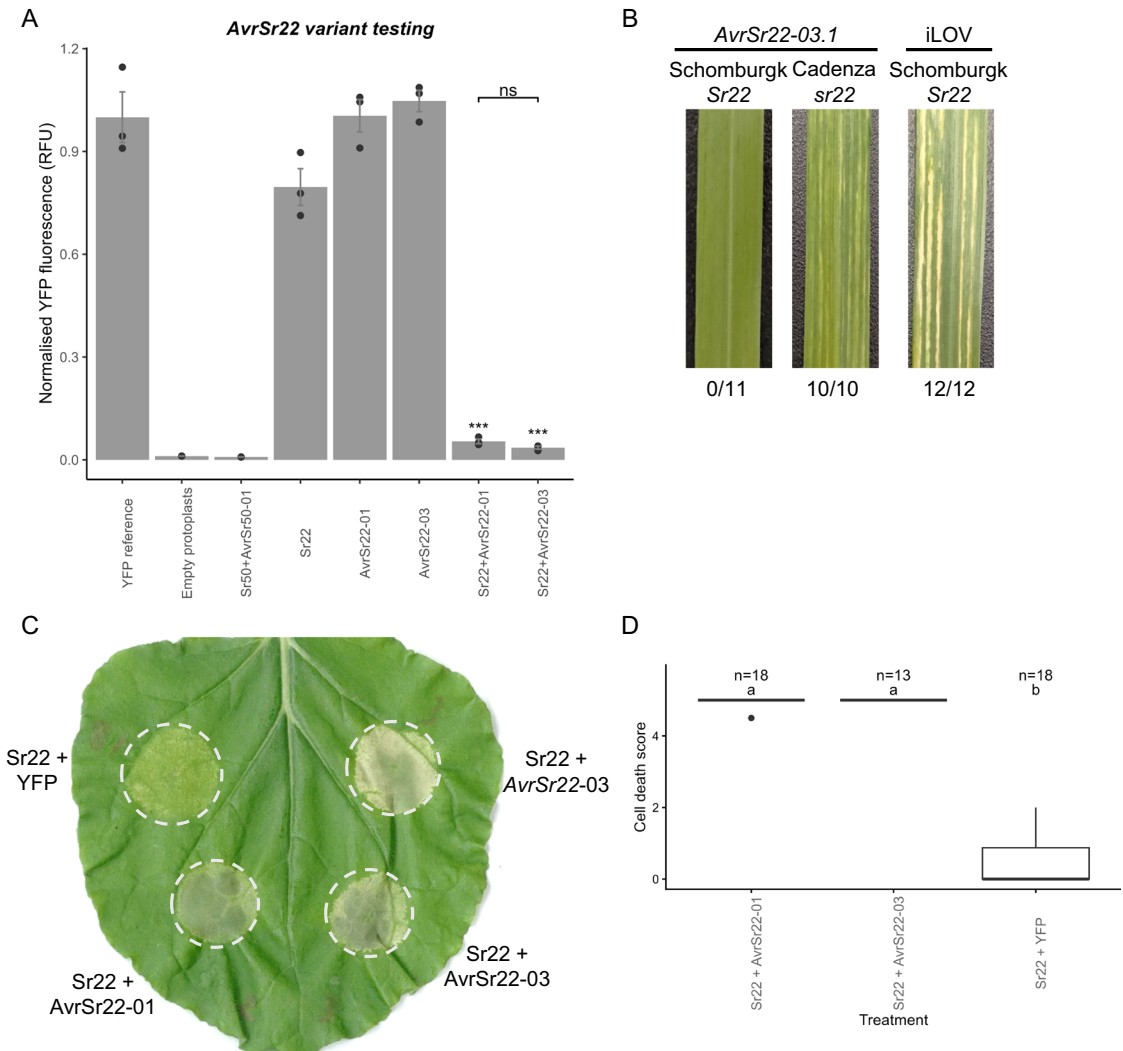

**Fig. 5 | Functional characterization of *AvrSr22-03*. A** Transient expression in protoplasts of the wheat cultivar Fielder. The *x*-axis shows genes used for transformation while the *y*-axis shows fluorescence values normalized to the YFP reference (RFU). Grey columns show mean values from three biological replicates. Error bars indicate the standard error. Black dots represent the result for each replicate of a given treatment. Two-tailed t-tests were performed for pairwise comparisons between specific treatments, '***' indicates significance at 0.001 level (*P*-value = 2.165e-4, 1.99e-4), 'ns' non-significant. A *t*-test significance value directly above a column indicates a comparison with the YFP reference. Raw data, detailed analysis and t-test *P*-values are provided in Source data file. **B** Transient expression of *AvrSr22-03* and negative control iLOV in wheat cultivars Schomburgk (*Sr22*) and Cadenza (*sr22*) using the BSMV-based protein over-expression system. Fractions under leaf pictures indicate the number of plants presenting Barley Stripe Mosaic Virus (BSMV) symptoms over the total number of inoculated plants. **C** Co-expression assay in *Nicotiana benthamiana*. Dotted lines mark out the area infiltrated with an *Agrobacterium tumefaciens* suspension. **D** Cell death score in all infiltrated *N. benthamiana* leaves represented as a box and whiskers plot with the number (n) of replicates indicated in the graph. The *x*-axis indicates the different treatments, the *y*-axis represents the cell death score out of 5 (0 = no cell death, 5 = cell death over the whole inoculated area). Letters indicate samples that do not differ significantly (Welch's ANOVA followed by Dunnett's T3 multiple comparisons test, pairwise *P*-values in Source Data File).

generated for the US isolate 99KS76[27], but this still contains phase swaps, with each of the two pseudo-haplotypes (designated E and F) containing a mixture of sequences derived from either nucleus (Henningsen et al., unpublished data). Previous SNPchip analysis of isolates derived from the 2013 stem rust outbreak in Ethiopia revealed the existence of two distinct lineages amongst isolates sharing the TKTTF pathotype, clades IV-A and IV–B[8]. Phylogenetic analysis placed isolate ETH2013-1 in clade IV-B. This clade also includes an isolate detected in the UK in 2013[30], suggesting the widespread dispersal of the clade IV-B TKTTF lineage prior to the Ethiopian outbreak in 2013, as observed for the clade IV-A lineage[2,28]. Genome sequence data confirms that the IV-A clade is genetically distinct to IV-B with no evidence of shared nuclear haplotypes, despite sharing the same race type (TKTTF). Other isolates collected in Germany and Georgia displayed similar pathotypes to TKTTF but SNPchip genotyping placed them into other distinct

genetic subgroups within clade IV (IV-C, IV-D, etc), and it was suggested that this set of lineages may derive from a sexually recombining population[2,31]. Genome assembly of additional isolates from these clade IV subgroups will help to resolve their genetic relationships and the evolutionary origins of this clade.

We previously showed that the Ug99 lineage arose via somatic hybridization and nuclear exchange, with hap01 donated by Pgt21-0 and hap03 by an unknown parent[22]. Although the IR-01 isolate contains the hap03 nuclear genome it was first isolated in 2015, well after the emergence of Ug99 lineage; thus, it was not clear whether it represented a parental or derivative lineage[22]. Guo et al.[27] proposed the clade II lineage, which was present in Africa prior to the Ug99 epidemic, as a possible donor of hap03. Here we found that IR-01 is a member of clade II and confirmed that these isolates do fully contain hap03, suggesting nuclear exchange between these lineages. However, the earliest

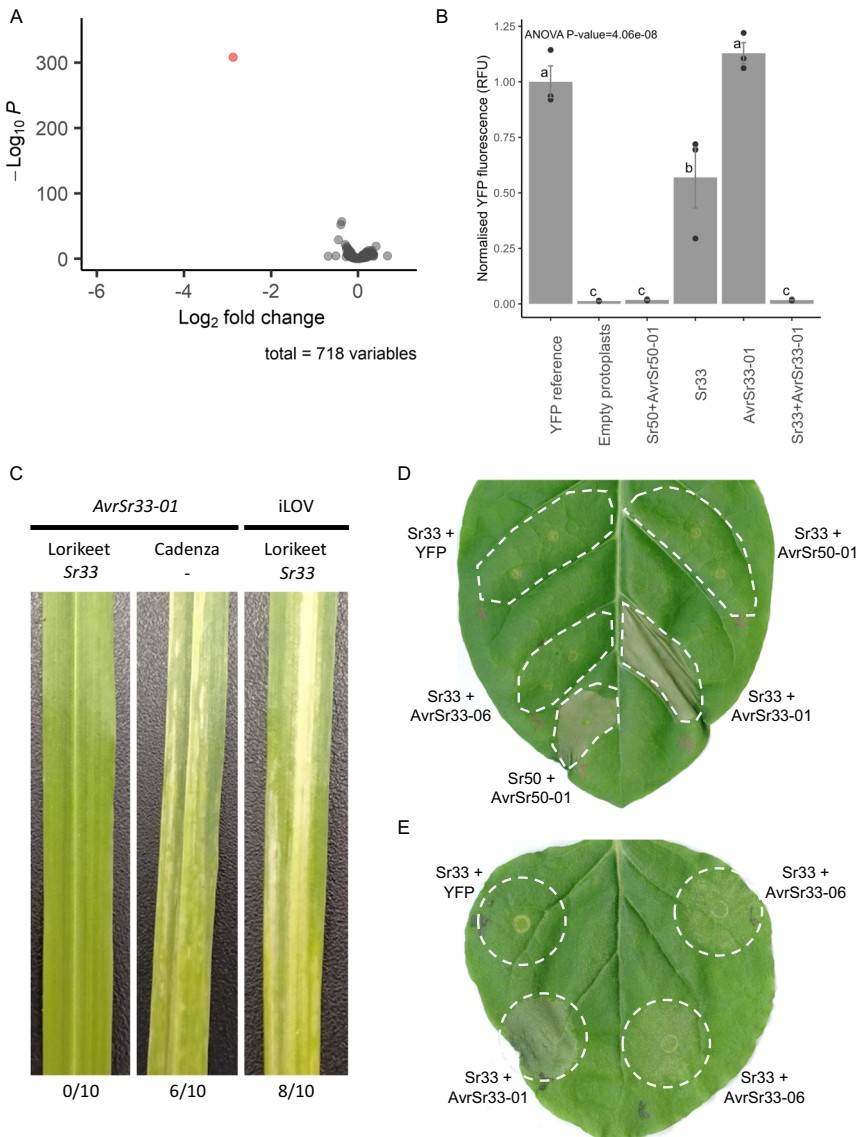

**Fig. 6 | Identification of *AvrSr33* by effector library screening. A** Differential gene expression analysis of a library comprising 1373 predicted effector genes from *P. graminis* f. sp. *tritici* (*Pgt*) divided into two pools, part A and part B, after co-expression in wheat protoplasts of cultivar Fielder with *Sr33* compared to an empty vector. The volcano plot shows differential expression (*x*-axis) versus adjusted *P*-value (*y*-axis) for each effector construct (dots). The effector gene construct that showed significantly reduced expression is represented in red. Shown here is the result for library part A (718 effector candidates). The result for part B (655 effector candidates) can be found in Supplementary Fig. 22. **B** Validation of the interaction between *Sr33* and *AvrSr33* in protoplasts of wheat cultivar Fielder. The *x*-axis represents the different treatments, the *y*-axis represents fluorescence normalized to the YFP reference (RFU). Grey columns show mean values from three biological replicates. Error bars indicate the standard error. Black dots represent the result for each replicate of a given treatment. Analysis of variance (ANOVA, two-tailed Tukey post hoc test) test detected significant differences between samples (*P*-value = 4.06E-08) and letters indicate samples that do not differ significantly (pairwise *P*-values in Source Data File). **C** Infection of wheat cultivars Lorikeet (*Sr33*) and Cadenza (*sr33*) with BSMV expressing *AvrSr33-01* or the negative control iLOV. Fractions under leaf pictures indicate the number of plants presenting Barley Stripe Mosaic Virus (BSMV) symptoms over the total number of inoculated plants. **D** Validation of the interaction between *Sr33* and *AvrSr33* by co-expression in *Nicotiana tabacum*. Dotted lines mark out the area infiltrated with an *Agrobacterium tumefaciens* suspension. **E** Validation of the interaction between *Sr33* and *AvrSr33* by co-expression in *N. benthamiana*. Dotted lines mark out the area infiltrated with an *A. tumefaciens* suspension.

collected isolate confirmed as part of the Ug99 lineage was from 1972 (72ETH11-4), while the earliest collected isolate placed in clade II was from 1984 (84KEN8B). The question of which is the parental or derived lineage remains open. We also found that hap05 in ETH2013-1 is fully contained in IR-06, which may indicate another nuclear exchange event. However, an alternative explanation is that hap05 is a direct sexual product of IR-06 that is fully derived from the two unknown IR-06 haplotypes by meiotic recombination (Supplementary Fig. 26). Henningsen et al.[36] found such a relationship between haplotypes in two lineages of the oat crown rust fungus (*Pca*) from Australia. These lineages showed a non-reciprocal haplotype-sharing arrangement in that one haplotype of lineage L11 (hap25) was fully contained in genomic sequences from lineage L18, but neither haplotype of L18 (hap03 and hap04) was fully contained in L11. Recombination block analysis of the haplotype-resolved genome assemblies revealed that hap25 was 50% derived from each of hap03 and hap04, consistent with it being the $F_1$ product of meiotic recombination in the L18 parent. Given the postulated sexual origin of the clade IV lineages[2], a similar relationship may exist between ETH2013-1 and IR-06. Preliminary data suggests that IR-06 belongs to clade IV-C and full genome assembly of

IR-06 or another member of this lineage will be required to answer this question. Isolates CZ-02,03 and PK01,02 represent two different clonal lineages that each share hap02 from Pgt21-0[22] (Supplementary Fig. 8B, Supplementary Data 10). Given the age of the Pgt21-0 lineage (>100 years), it is most likely that these lineages acquired hap02 as a result of nuclear exchange. Again, full genome assembly of isolates from these lineages would clarify this relationship.

Like other *Puccinia* species[22,36,38,40,45], *Pgt* seems to contain a bi-allelic *a* (or *PR*) mating locus and multi-allelic *b* (or *HD*) locus with eight alleles now defined at the sequence level. All 146 *Pgt* isolates with available genome sequence data show heterozygosity at the *a* mating locus (*STE3.2.2/STE3.2.3*). While the *b* locus alleles of the seven reference haplotypes are all distinct, this is not sufficient to distinguish them from other haplotypes. For example, isolates IT-05, IR-06, IR-02 and SC-01 all share the *bW6bE6* allele with hap05 of ETH2013-1 but only IR-06 contains hap05.

The identification of *Avr* genes from *Pgt* has been accelerating with the emergence of new screening tools[15,16]. Here we identified *AvrSr33* in a protoplast screen from a library of 1,373 *Pgt* effector candidates and confirmed its recognition using VOX assays in wheat and co-expression with *Sr33* in *N. benthamiana*. This brings the total number of *Avr* loci identified in *Pgt* to seven. The fully haplotype-phased genome references generated here provide an ideal resource for assessing genetic variation at these loci. We identified the genotypes of the seven *Avr* loci in each of the *Pgt* reference haplotypes and functionally characterized 22 novel *Avr* gene variants for recognition by the corresponding *Sr* genes. This allowed assignment of *Avr* genotypes to these epidemiologically important lineages of the wheat stem rust pathogen which can explain their virulence phenotypes on wheat lines carrying the corresponding *Sr* genes. Isolates of the TTRTF race from Italy were previously reported to be virulent on *Sr13b*, but avirulent on a wheat differential line (ST464) containing *Sr13c*[2,9,32]. However, we found that in addition to being virulent on *Sr13b*, ITA2018-1 was also virulent on a transgenic Fielder line containing *Sr13c*, which is consistent with the homozygous deletion of *AvrSr13* in this strain. This suggests that the ST464 line may contain an additional *Sr* gene (or genes) effective against this strain, possibly explaining some of the reported differences in recognition specificity between *Sr13b* and *Sr13c*, which encode NLR receptor proteins that differ by only two amino acids. This highlights the value of using near-isogenic wheat lines carrying single *R* genes for accurate assignment of pathogen virulence phenotypes. ITA2018-1 was also virulent on transgenic wheat line Fielder expressing a *Sr35* and is homozygous for the novel *avrsr35-02* allele, which was not recognized by *Sr35* in co-expression assays.

Apart from *AvrSr13* and *AvrSr35* in ITA2018-1 each of the four reference genomes contains at least one recognized allele at each of the *Avr* loci, consistent with their avirulent phenotypes on wheat lines carrying the corresponding *Sr* genes. As noted by Chen et al.[20], *Avr62-05* and *-07* gave strong recognition responses in functional assays, and we found similar strong responses for the closely related variants *Avr62-08* and *-11*. However, *AvrSr62-01* and *-04* gave intermediate recognition responses[20], so it is still not clear if hap01 and hap05 containing only these two recognized variants express an avirulent phenotype on *Sr62* wheat. *AvrSr22* was the only locus at which all the isolates were homozygous for strongly recognized alleles. This means that evolution of virulence to *Sr22* in these lineages would require two independent mutations. Thus, *Sr22* may provide more durable resistance against these *Pgt* strains.

Our findings indicate that it is critical to detect nuclear haplotype exchanges as these will impact the overall evolution of the pathogen and virulence profiles of rust populations. Given that avirulence is a dominant trait in rust fungi, nuclear exchanges may be unnoticed by rust surveillance practices that rely solely on virulence phenotyping across the relatively small wheat differential set. This is exemplified by the presence of Pgt21-0 in Australia since the 1950's which carries a nuclear haplotype present in the Ug99 lineage[22]. Characterizing the genetic diversity of *Avr* genes among major *Pgt* races causing epidemics is important for monitoring rust evolution and prioritizing resistance breeding strategies. The catalogue of *Avr* genes and recognition properties provides insight into the virulence mechanisms behind major disease outbreaks, such as the deletion of *AvrSr13*. This change allowed the TTRTF race group to infect durum wheat lines containing *Sr13b*, which were prevalent in the field during the outbreak in Sicily in 2016[9]. Continued development of a comprehensive *Avr* gene atlas with additional reference haplotypes and new *Avr* genes will provide a valuable resource to enable virulence prediction from sequence-based surveillance tools such as the MARPLE platform[46,47].

## Methods

### Wheat stem rust isolate sampling and virulence phenotyping
The *Pgt* isolate ETH2013-1 was collected from the Bale region of Ethiopia in 2013 from the bread wheat (*Triticum aestivum* L.) cultivar (cv) Digalu. Isolate ITA2018-1 was collected from Ciminna, Sicily, Italy in 2018 from the bread wheat cv Antonello. Urediniospores of ETH2013-1 and leaf samples infected with ITA2018-1 were sent to the University of Minnesota and processed within the Biosafety Level 3 facility (St Paul, MN, USA). Single pustules of each isolate were selected from the susceptible wheat cv McNair 701 and increased several times to obtain pure cultures as described by Li et al.[22]. Isolates were typed for race based on the infection types they exhibited on the wheat stem rust standard differential set[48] (Supplementary Data 1). Infection types ranging from 0 to 2 and combinations thereof were indicative of low compatibility (avirulent) and those from 3 to 4 were indicative of high compatibility (virulent) based on the 0 to 4 scale of Stakman et al.[49]. Isolates were also phenotyped for their infection types on transgenic wheat lines of cv Fielder containing the single genes *Sr13c*, *Sr22*, *Sr27*, *Sr33*, *Sr35*, and *Sr50*. Additionally, isolates ETH2013-1 and ITA2018-1 were assayed for their infection types on the wheat introgression line Zahir 1644, containing *Sr62* derived from the wild wheat *Aegilops sharonensis*[20] (Supplementary Data 1). To generate the *Sr13c* transgenic line, a 9,016 bp genomic sequence (NCBI accession number KY924305.1) comprising 2,913 bp upstream and 2,590 bp downstream of the *Sr13c* coding sequence was synthesized (Epoch, USA) and cloned into pD12 donor vector as previously described in ref. 50. The *Sr13c* gene was transferred into a binary vector with Bar selection using Gateway™ LR Clonase™ Enzyme mix (Invitrogen, USA) and transformed into wheat cv Fielder as previously described in ref. 51.

### Genome sequencing
DNA for short read sequencing was extracted from 30 mg of urediniospores using the Omniprep™ Genomic DNA isolation kit (#786-136, G-Biosciences), according to the manufacturer's instructions. Samples were sequenced by the University of Minnesota Genomics Center (UMGC, St. Paul, MN) using 150 bp paired-end Nextera Flex library preparation and the Illumina NovaSeq 6000 platform. High molecular weight DNA for long-read sequencing was extracted from ~450 mg of urediniospores as described by Li et al.[22] and sequenced on the PacBio Sequel II platform at UMGC. For Hi-C sequencing of ETH2013-1, ITA2018-1, and Ug99, 350 mg of urediniospores were cross-linked in 1% formaldehyde (20 min at ~22 °C) and then glycine was added to a 125 mM final concentration. After 15 min incubation, urediniospores were washed twice with distilled $H_2O$, centrifuged at 3500 g for 5 min, and the supernatant removed. Washed urediniospores were ground in liquid $N_2$ and shipped on dry ice to Phase Genomics Inc. (Seattle, WA) for Hi-C library preparation. Paired-end libraries were sequenced with Illumina NovaSeq at Azenta Life Sciences (South Plainfield, NJ).

### RNA sequencing
Susceptible wheat plants (cv McNair 701) were inoculated with freshly collected urediniospores and maintained as above. At five- and seven-

days post inoculation, infected leaves were sampled in triplicate and frozen in liquid $N_2$. Germinated urediniospore samples were prepared by dusting 200 mg of fresh spores over a germination solution as previously described in ref. 52, collected after 10 h, and frozen in liquid $N_2$. All tissues were ground to a fine powder using a mortar and pestle in liquid $N_2$ and the RNA extracted using the Qiagen RNeasy Plant Mini kit (Cat No: 74904). Samples were sequenced at UMGC on the Aviti Cloudbreak platform after preparation of 150 bp paired-end stranded mRNA libraries (high output sequencing kit).

## Genome assembly and scaffolding

PacBio HiFi reads were assembled using hifiasm[53] v0.19.8 in Hi-C-integration mode using corresponding Hi-C reads with default parameters. The initial assembly was cleaned as described in Sperschneider et al.[38] and Henningsen et al.[36]. Briefly, HiFi reads were aligned to the assembly using minimap2[54] v2.26 and coverage was assessed using the pileup.sh tool from bbmap v39.01 (https://sourceforge.net/projects/bbmap/). Low coverage contigs were removed from assemblies after determining cut-offs (7x for ETH2013-1, 10x for ITA2018-1) with a custom R script (https://github.com/JanaSperschneider/GenomeAssemblyTools/tree/master/CollapsedGenomicRegions). Mitochondrial and contaminant contigs were identified using BLASTn[55] and removed from the assemblies. The NuclearPhaser v1.1 pipeline was used to assess phasing (https://github.com/JanaSperschneider/NuclearPhaser). Scaffolding was performed using information from SALSA[56] v2.2, Hi-C-Pro[57] v3.1.0, Hicexplorer[58], and synteny to hap01 from isolate Pgt21-0 (GenBank accession GCA_008522505.1) assessed by alignment with D-GENIES[59]. BUSCO3[60] v3.0.2 was used to assess completeness of the assemblies (-l basidiomycota_odb9). Telomeres were detected using the findtelomeres.py script (https://github.com/JanaSperschneider/FindTelomeres). The published Ug99 contigs[22] were also phased and scaffolded into chromosomes as described above.

## Gene annotation

Genes and repeats were annotated in the ETH2013-1 and ITA2018-1 genomes as described by Sperschneider et al.[38] and Henningsen et al.[36]. Briefly, RepeatModeler v2.0.2a (-LTRStruct) and RepeatMasker v4.1.2p1 were run to model and mask repeats in the genomes (http://www.repeatmasker.org). The full Repeatmodeler library was used to mask the genomes for statistics, whereas for gene prediction a library of only classified repeat families was used. RNAseq reads were aligned to their respective genomes with HISAT2[61] v2.2.1 and transcripts assembled with genome-guided Trinity[62] v2.13.2 and Stringtie[63] v2.2.1. CodingQuarry[64] v2.0 was run in pathogen mode on the transcripts. We then ran funannotate[65] v1.8.5 predict and supplied Trinity transcripts, Coding-Quarry predictions, and Pucciniomycotina EST clusters from JGI Myco-Cosm (http://genome.jgi.doe.gov/pucciniomycotina/pucciniomycotina.info.html). To capture additional genes encoding secreted proteins, we ran TransDecoder v5.5.0 (https://github.com/TransDecoder/TransDecoder) on the StringTie infection transcripts and selected complete ORFs encoding proteins with a signal peptide predicted by SignalP[66] v4.1 and no transmembrane domains predicted by TMHMM[67] v2.0. Novel secreted protein encoding genes were added to the annotation unless they were >90% contained in a repetitive region.

## Whole genome comparison and k-mer containment analysis

Whole genome alignment was performed with MUMmer[68] v4.0.0rc1 (default nucmer parameters), and sequence divergence was assessed with dnadiff[68]. Synteny plots of *Pgt* chromosomes were generated with MUMmer[68] v4.0.0rc1 (nucmer --maxmatch -c 100 -b 500 -l 50; delta-filter -m -i 60 -l 1000; show-coords -THrd), syri[69] v1.6.3 and plotsr[70] v1.1.0. The long arm of chromosome 3 in hap01 from isolate Pgt21-0 in NCBI was found to have an incorrect inversion from a scaffolding gap (position = 4,014,136 bp) to the end of the chromosome, which was corrected for the synteny visualization. Recombination analysis was performed using the pangenome-graph approach described by Henningsen et al.[36] using cactus[71] v2.9.0. Mash[72] v2.0 was used to measure k-mer containment of sketched reference haplotype-phased sequences (-s 100000) in genomic Illumina reads from *Pgt* isolates. Isolates with k-mer identity of ≥ 99.85% and shared k-mers ≥ 97.50% were considered as likely containing the screened haplotype.

## Phylogenetic analysis

In addition to the Illumina data generated for isolates ETH2013-1 and ITA2018-1, we also utilized publicly available RNAseq and genomic data from 151 *Pgt* isolates[18,19,22,27,30,35,73–75] (Supplementary Data 13). Genomic data for seven isolates were excluded because SNP allele frequency and k-mer containment analysis showed signs of sample contamination (Supplementary Data 13). Illumina reads were processed using fastp[76] v.0.23.2. Genomic reads were aligned to the haplotype-phased reference genomes of isolates Pgt21-0, Ug99[22], ETH2013-1, and ITA2018-1 using bwa-mem2[77] v.2.2.1. RNAseq reads were aligned to the same references with STAR[78] v.2.7.9a. Samtools[79] was used to prepare alignment files for variant calling with Freebayes[80] v1.3.6 (--use-best-n-alleles 6). Variants were filtered with vcflib[81] v1.0.1 (-f "QUAL > 20 & QUAL / AO > 10 & SAF > 0 & SAR > 0 & RPR > 1 & RPL > 1 & AC > 0") and vcftools[82] v0.1.16 (--min-alleles 2 --max-alleles 2 --max-missing 0.9 --maf 0.05). The vcf file was converted to Phylip format using vcf2phylip (https://github.com/edgardomortiz/vcf2phylip/blob/master/vcf2phylip.py). Maximum likelihood tree search and bootstrapping were conducted with RAxML[83] v8.2.12 (options -f -a -m GTRCAT -p 12345 -x 12345 -#500 –no-bfgs).

## Assessment of mating type loci

To identify putative mating loci in isolates ETH2013-1 and ITA2018, protein sequences of *STE3.2.2*, *STE3.2.3*, *bW-HD1*, and *bE-HD2* from Pgt21-0[22] were BLASTed (tblastn) against genome references with BLAST + [55] v2.13.0. Proteins overlapping the best hits were examined for conserved domains and intron/exon count[22,40,84]. Protein sequences for *bW-HD1* and *bE-HD2* alleles (Supplementary Data 8) were aligned with CLUSTALW[85], and phylogenetic trees were generated with the 'raxml-bootstrap' option. Protein trees were rooted on alleles from *Pca* isolate Pca203 (*Pca bW1* and *Pca bE1*)[24] with R package 'ape'[86] and visualized with iTOL[87]. To detect *STE3.2* and *HD* alleles in *Pgt* isolates, the coding sequences of individual exons were extracted, sketched together with mash[72] v2.0 (-s 1000), and then screened with whole genome or RNAseq Illumina reads for 146 *Pgt* isolates (Supplementary Data 13). The *b* locus alleles were considered contained at or above 99% k-mer identity, and 90% shared k-mers.

## High throughput effector screening in wheat protoplasts

A library of 1,373 effector candidates from *Pgt*[15,20] was screened in two pools (part A and part B, containing 718 and 655 clones, respectively). Pooled libraries were propagated in ElectroMAX™ Stbl4™ competent cells, and DNA was isolated using the QIAGEN EndoFree Plasmid Giga kit. The reporter (pTA22-*YFP*), library proxy (pTA22-*AvrSr50*), empty vector (pTA22), controls pTA22-*Sr50* and pTA22-*Sr35*, and pTA22-*Sr33* plasmids were isolated from *E. coli* DH5α using the Macherey Nagel NucleoBond Xtra Midi Plus EF kit. A total of 300,000 protoplast cells of wheat cv Fielder were transformed with a pooled effector library at a multiplicity of transfection (MOT) of 0.14 million molecules per cell (mmpc) for each construct, corresponding to 162 μg and 163 μg for parts A and B of the library, respectively, along with either the empty vector pTA22 (MOT = 36mmpc, 54 μg), controls pTA22-*Sr50* (MOT = 36mmpc, 86 μg) and pTA22-*Sr35* (MOT = 7mmpc, 17 μg), or pTA22-*Sr33* (MOT = 36mmpc, 100 μg) in triplicate as described in ref. 15. pTA22-*Sr35* was used at a lower MOT to reduce cell death caused by auto-activity of the Sr35 protein in protoplasts. A control protoplast sample (300,000 cells) was transformed with pTA22-*YFP* (MOT = 36mmpc, 62 μg) and pTA22-*AvrSr50* (MOT = 96mmpc, 157 μg) as a library proxy to assess transfection efficiency by fluorescence as above.

Total RNA was extracted from transformed protoplasts after 20 hours using the Maxwell® RSC Plant RNA Kit (Cat. #AS1500) with the Maxwell® RSC 48 Instrument (Cat. #AS8500) following the manufacturer's protocol with minor modifications. Briefly, 500 μL of homogenization buffer was added to each sample along with 250 μL of lysis buffer and then 600 μL of the cleared lysate was transferred to a Maxwell® RSC Cartridge for RNA extraction. The RNA was eluted in 55 μL of nuclease-free water. Library-specific cDNA synthesis and PCR was carried out on 1 μg RNA using the Invitrogen SuperScript IV One-Step RT-PCR System with ezDNase kit (Cat. # 12595100) with the forward and reverse primers ZmUbi1_5UTR_F3b (5′-GCACACACACACAACCAG-3′) and FS_cDNA_R (5′-TGCTAGATCTCGACAGTACG-3′) as previously described in ref. 15. The cDNA was sent to the ACRF Biomolecular Resource Facility, The John Curtin School of Medical Research, Australian National University, for Illumina library preparation and sequencing on the NovaSeq X Series with a 1.5B flow cell with 100 bp single-end reads.

Sequencing reads were cleaned using fastp[76] v0.23.2. The clean reads were aligned to the coding sequences of the effector candidates with STAR[78] 2.7.9 (--alignIntronMax 1, --outFilterMultimapNmax 10). Read counts were compiled with samtools[79] idxstats and imported into DESeq2[88] for differential expression analysis using default parameters followed by lfcShrink (type = "apeglm") to compare the *Sr* and empty vector treatments. Volcano plots were produced with EnhancedVolcano (https://github.com/kevinblighe/EnhancedVolcano). Syntenic regions in the *Pgt* genome assemblies were called with minimap2[54] (-X -N 50 -p 0.1 -c) and plots were produced with gggenomes (https://github.com/thackl/gggenomes).

### Identification of *Avr* gene alleles

*Avr* gene alleles were identified by a tblastn search (NCBI toolkit version 25.2.0) with the phased genome sequences as query sequences against a local protein sequence database comprised of AvrSr13 (PGT21_021053), AvrSr22 (PGT21_017626), AvrSr27 (PGT21_006532), AvrSr35 (PGT21_019625) AvrSr50 (PGT21_020314), AvrSr33 and AvrSr62 (PGT21_030464, PGT21_030510, PGT21_030812, PGT21_030832, PGT21_036621, PGT21_036648, PGT21_036655). The genomic coordinates of the best matches were used to extract the corresponding *Avr* coding sequences using samtools[79] faidx. Exon/intron boundaries were established from RNAseq data aligned to the assemblies using TopHat2[89]. Protein sequences were aligned with ClustalW[85] v2.1 and ML trees were generated with the 'raxml-bootstrap' option. Synteny plots were generated with R package gggenomes[90] v1.0.1 using alignments generated with minimap2[54] v2.25 (options -X -N 1 -g 300 -m 20 -O7 -E7).

### Transient expression in *Nicotiana benthamiana* and wheat protoplasts

The coding sequences (excluding signal peptides) of *Avr* gene alleles were codon optimized for wheat and synthesised either directly in pDONR207 (Epoch Life Science) or as G-blocks with Gateway overhangs (Integrated DNA Technologies) and cloned into pDONR207 and subsequently into expression vectors via a Gateway BP and LR reactions (Invitrogen) respectively. Amino acid mutations were introduced by Dpn1-mediated site-directed mutagenesis using Phusion High-Fidelity DNA polymerase (NEB, Cat. #M0530L). Plasmids used in this study are listed in Supplementary Data 14. *Nicotiana benthamiana* and *N. tabacum* plants were grown in a growth chamber at 23 °C with a 16 h light period and used for agroinfiltration at the 4-week-old stage. *Agrobacterium tumefaciens* cultures of strains GV3103-pMP90RK (pAM constructs) and GV3101-pMP90RK (pBIN19 constructs) were grown at 28 °C overnight with shaking at 200 RPM in Luria-Bertani liquid medium with appropriate antibiotic selections. Cells were pelleted by centrifugation and resuspended to a final concentration of $OD_{600} = 0.2$ for the *Agrobacterium* strains carrying R genes and $OD_{600} = 0.5$ for those carrying Avr genes in 10 mM MES pH 5.6, 10 mM $MgCl_2$, 150 μM acetosyringone and

incubated at 20–23.5 °C for 2 h before infiltration. Leaves were photographed or scanned 3 to 5 days after infiltration.

For protoplast assays, plasmids were extracted using the NucleoBond Xtra Midi EF kit (Macherey-Nagel). Wheat seedlings (cv Fielder) were grown in Martins Seed Raising and Cutting Mix at 24 °C under a 12 h light (~130 μmol m$^{-2}$ s$^{-1}$) and 12 h dark cycle. At seven days, the first leaf was harvested for protoplast isolation and cells transformed as described in ref. 15. Following a 20 h incubation, protoplasts were transferred to 2 mL tubes and left to settle for 30 m. Half of the volume of supernatant was removed and the protoplasts resuspended in the remainder. 130 μL of each sample was transferred in duplicate to a Black 96-Well Immuno Plate (Thermofisher, Cat. #9502867). YFP fluorescence was measured using a CLARIOstar Plus plate reader (BMG Labtech; the top optic was used, optic settings: presetname = YFP, excitation = 497-15, dichroic filter = 517.2, emission = 540-20, focal height = 5.7 mm, scan settings: number of flashes per scan = 4, scan mode = matrix scan, scan matrix dimension = 30 x 30, scan width = 7 mm). The gain was set based on fluorescence scanning of YFP control samples (scan settings as above except: number of flashes per scan = 100, scan mode = spiral average). Fluorescence was normalized against the mean of the YFP control samples for each experiment.

### Virus-mediated *in planta* effector expression

We employed the Barley Stripe Mosaic Virus (BSMV)-based protein overexpression system comprising three T-DNA binary plasmids: pCaBS-α, pCaBS-β, and pCa-γb2A-LIC[19,91]. Coding sequences of mature Avr protein variants (no SP) were synthesized (Twist Biosciences, South San Francisco, CA, USA), PCR-amplified using primers listed in Supplementary Data 15 and then inserted into the pCa-γb2A-LIC vector. BSMV:iLOV was used as a negative control[92,93]. Each of the three BSMV binary vectors was introduced into *A. tumefaciens* strain GV3101 (pMP90) and *N. benthamiana* plants (3-4 weeks old) were agro-infiltrated as previously described[91]. Infected tissue was used to prepare sap for mechanical inoculation of wheat plants carrying the appropriate *Sr* resistance genes (Supplementary Data 16) at the three-leaf stage. Inoculated wheat plants were maintained under controlled growth conditions with day/night temperatures of 23 °C/20 °C, 60-65% relative humidity, a 16 h photoperiod, and light intensity of approximately 220 μmol m$^{-2}$ s$^{-1}$. *AvrSr13* assays were conducted at slightly elevated day/night temperatures of 25 °C/22 °C, as *Sr13* resistance is more effective at higher temperature[94]. Systemic leaves were inspected at 10,16 and 24 days post-inoculation, and the presence or absence of viral symptoms was recorded. For each construct, at least ten plants were tested per experiment, with all experiments independently repeated and yielding consistent results.

### Reporting summary

Further information on research design is available in the Nature Portfolio Reporting Summary linked to this article.

## Data availability

All raw sequence data are available under NCBI BioProject PRJNA1267768. Assembly and annotation files are deposited in the CSIRO data access portal (https://data.csiro.au/collection/csiro:65828). Source data are provided with this paper.

## Code availability

Scripts are available at https://github.com/henni164/epidemic_pgt (Zenodo https://doi.org/10.5281/zenodo.18322261).

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

## Acknowledgements

This project was supported by USDA-NIFA award 2022-67013-36505 to BJS and BBSRC grant BB/W018403/1 to KK as part of the NSF/BBSRC Lead Agency Agreement, the CSIRO Research Office to JS, PND and MF (OD-227545, OD-235285), the CSIRO Synthetic Biology Future Science Platform to TV (OD-213047), the Grains Research and Development Corporation project CSP2403-014RTX to PND, the Lieberman-Okinow Endowment at the University of Minnesota to BJS, the 2Blades Foundation to MF and the Gatsby Foundation to PND, MA, MF. ECH was supported by an ANU University Research Scholarship and an ANU/CSIRO Digital Agriculture PhD Supplementary Scholarship. We acknowledge Dr David Hodson at CIMMYT for providing the rust isolate ETH2013-1, as well as Dr Matthew J Moscou and Kim-Phuong Nguyen for discussions and feedback as well as technical support. We truly appreciate their unwavering support. We thank Stephanie Dahl and Aubree Kees for their assistance at the University of Minnesota Biosafety Level-3 containment facility, the Minnesota Supercomputing Institute, and CSIRO High Performance Computing Services for computational resources, and Peter Tyson and Joel Hansen for providing computational support. We thank Biagio Randazzo (AS.A.R. - Società Semplice Agricola Randazzo, Baucina (PA), Italy) and Massimo Palumbo (CREA - Council for Agricultural Research and Economics, Research Center for Cereal and Industrial Crops, Acireale (CT), Italy) for their involvement in processing the rust isolate ITA2018-1.

## Author contributions

Conceptualization: K.K., P.N.D., B.J.S., M.F.; Data curation: R.S., E.C.H., C.L.P., J.S.; Formal analysis: R.S., E.C.H., C.L.P., J.C., J.L., O.M., J.S.; Investigation: R.S., E.C.H., C.L.P., J.C., J.L., O.M., D.L., L.C.C., Z.S., A.F., E.S.N., F.L., M.A.O.; Project Administration: R.S., C.L.P., P.N.D., B.J.S., M.F.; Software: R.S., E.C.H., J.S.; Methodology: E.C.H., C.L.P., J.L., M.A.O., T.A., T.V., K.K., J.S., P.N.D., B.J.S., M.F.; Visualization: R.S., E.C.H., C.L.P., J.C., J.L., O.M.; Validation: C.L.P., J.C.; Writing – original draft: R.S., E.C.H., C.L.P., J.S., P.N.D., M.F.; Writing – review & editing: R.S., E.C.H., C.L.P., J.C., J.L., O.M., D.L., L.C.C., Z.S., A.F., E.S.N., F.L., M.A.O., T.A., T.V., N.V., M.L., M.A., E.S., K.K., J.S., P.N.D., B.J.S., M.F.; Resources: T.A., T.V., N.V., M.L., M.A., B.J.S.; Supervision: C.L.P., J.C., M.A.O., T.V., E.S., K.K., J.S., P.N.D., B.J.S., M.F.; Funding Acquisition: E.S., K.K., P.N.D., B.J.S., M.F.

## Funding

## Competing interests

T.A, T.V., P.N.D., M.F., and J.S. are inventors on patent application WO2024103117 filed by CSIRO and relating to the identification of protein-protein interactions by protoplast screening. The remaining authors declare that they have no competing interests.
