## [Transparent Peer Review file · Nature Communications]

Allelic variation of Avr genes in highly virulent strains explains severe wheat stem rust epidemics

Corresponding Author: Dr Melania Figueroa

Version 0:

Reviewer comments:

Reviewer #1

(Remarks to the Author)

Spanner et al. present the generation of new phased genome assemblies for the wheat stem rust pathogen *Puccinia graminis* f. sp. *tritici*, which were subsequently leveraged to characterize previously unknown nuclear haplotypes. The authors found substantial diversity at previously known effector loci. The cell death inducing function of novel alleles discovered were determined and a novel effector locus (AvrSr33) was identified. Overall, the research presented here was very impressive and provided important new insights into Pgt genomics and host-pathogen interactions. The manuscript clearly demonstrates how allelic diversity and structural variation at Avr loci control virulence phenotypes and used cutting edge techniques for characterizing effector function in an experimental system that is typically difficult to perform functional validation studies. The resources generated here will also be extremely valuable for the rust genomics research community. The experiments and analyses were properly designed and executed and the conclusions are well supported. I really enjoyed reading this manuscript and only have a few minor suggestions for the authors' consideration.

Supp. Fig. 6: I am having some trouble understanding this figure. I'm assuming the % covered represents shared recombination blocks, but how do you interpret this based on the hierarchy? I suggest clarifying the text in the manuscript and figure caption. Also, although these shared blocks are a relatively small proportion of the whole genome, do any of these correspond to Avr loci or other notable genes?

L211-213: Is this shown in any of the phylogenies presented?

L239-260: It appeared that some isolates were not assigned to haplotypes via k-mer containment. Can this be interpreted as the existence of additional haplotypes? If so, this could be mentioned/discussed further.

L311-314: What are the nonsynonymous changes present in the AvrSr35-02 allele? It think it would be worthwhile to mention the specific amino acid changes and comment on how those specific mutations may lead to the observed lack of recognition.

L676: What *Agrobacterium* strain was used?

Reviewer #2

(Remarks to the Author)

The manuscript "Allelic variation of Avr genes explains virulence of wheat stem rust isolates associated with severe disease epidemics" provides chromosome-level and nuclear-phased genomes of two strains of *Puccinia graminis* f. sp. *tritici* (Pgt). These enabled the identification of two additional nuclear haplotypes, and the characterization of Avr variants. The work is very nicely written, provides very good resources and advances our understanding on Pgt virulence and evolution. I have very few and minor comments:

I would like to ask the authors to highlight more the novelty of the work. Several of the Avrs the authors characterized have already been identified. nuclear level sequencing was already obtained for Pgt. An additional Avr was cloned, but there are 6 that have already been cloned.

How did the full genome sequencing of the isolates ITA2018-1 and helped the cloning of AvrSr33?

Is there genetic exchange between the 2 nuclei of an isolate?

Version 1:

Reviewer comments:

Reviewer #2

(Remarks to the Author)

The authors addressed all my comments. Congratulations for the great work!

Re: **Decision on Nature Communications manuscript NCOMMS-25-6563**

Reviewer #1 (Remarks to the Author):

Spanner et al. present the generation of new phased genome assemblies for the wheat stem rust pathogen *Puccinia graminis* f. sp. *tritici*, which were subsequently leveraged to characterize previously unknown nuclear haplotypes. The authors found substantial diversity at previously known effector loci. The cell death inducing function of novel alleles discovered were determined and a novel effector locus (*AvrSr33*) was identified. Overall, the research presented here was very impressive and provided important new insights into Pgt genomics and host-pathogen interactions. The manuscript clearly demonstrates how allelic diversity and structural variation at *Avr* loci control virulence phenotypes and used cutting edge techniques for characterizing effector function in an experimental system that is typically difficult to perform functional validation studies. The resources generated here will also be extremely valuable for the rust genomics research community. The experiments and analyses were properly designed and executed and the conclusions are well supported. I really enjoyed reading this manuscript and only have a few minor suggestions for the authors' consideration.

Thank you very much for taking the time to review our paper, we appreciate your feedback.

Supp. Fig. 6: I am having some trouble understanding this figure. I'm assuming the % covered represents shared recombination blocks, but how do you interpret this based on the hierarchy? I suggest clarifying the text in the manuscript and figure caption. Also, although these shared blocks are a relatively small proportion of the whole genome, do any of these correspond to *Avr* loci or other notable genes?

The % covered represents the % of identity shared with all the haplotypes placed above, so the numbers would vary depending on placement of isolates in the list. Table S7 provides the numbers for each pairwise comparison. The only pair with any significant conserved blocks are hap05 and hap07 with 9.5 to 10.5% of their genomes shared (depends on the direction of comparison). In the order shown in the figure, hap05 only has ~3.5% shared blocks with the upper haplotypes (hap01 to 04), while hap07 had 10.5% shared with the above haplotypes (01 to 06). However, the shared regions with hap05 are indicated by the coloured segments in the hap07 genome that correspond to hap05.

None of the characterised *Avr* loci are located within regions common to the haplotypes included in this study.

L211-213: Is this shown in any of the phylogenies presented?

An additional panel (C) has been added to Supplementary Figure 7 indicating which Clade I isolates have the Ug99 *STE3.2.3* variant and which have the Pgt21-0 *STE3.2.3* variant. Virulence/avirulence to *Sr31* is also indicated in this panel.

The text has been amended to refer to the new figure panel:

L208-211: “Several clade I isolates share this mutation (04KEN15604, 06KEN19V3, 07KEN24-4, 14KEN58-1, Pgt-60, ET-01, and ET-02), although some others (Pgt-55, Pgt-59, Pgt-61, 72ETH11-4; **Supplemental Table 9**) do not, suggesting a recent mutation event in this lineage (**Supplemental Figure 7C**).”

L239-260: It appeared that some isolates were not assigned to haplotypes via k-mer containment. Can this be interpreted as the existence of additional haplotypes? If so, this could be mentioned/discussed further.

When an isolate is not assigned to a haplotype (or indeed two haplotypes) by k-mer containment, this would be because it contains another as yet undefined haplotype. We worked with seven haplotypes in this paper, our results show that stem rust haplotype diversity goes well beyond that number. Indeed, we have recently assembled full genomes for a number of additional isolates and have identified about 25 unique haplotypes. This will be captured in a subsequent publication.

L311-314: What are the nonsynonymous changes present in the AvrSr35-02 allele? It think it would be worthwhile to mention the specific amino acid changes and comment on how those specific mutations may lead to the observed lack of recognition.

There are 40 single amino acid differences and 2 indels (1 aa and 4 aa) between AvrSr35-01 and AvrSr35-02, but only one of these differences lies in the interaction interface with Sr35 characterized by Zhao et al. (2022) and Förderer et al. (2022), S349 (AvrSr35-01) to R352 (AvrSr35-02). Our experimental data in *N. benthamiana* shows that this single mutation is sufficient to determine the difference in recognition of these two variants.

We have added two panels Supplementary Figure 21 presenting the effect on the interaction with Sr35 of S349R and R352S mutations in AvrSr35-01 and -02 respectively in *N. benthamiana*. The plasmids used have been added to Supplemental table 14.

We have amended the text as follows:

L311-318

“The AvrSr35-02 protein failed to trigger cell death when co-expressed with Sr35 in *N. benthamiana* and induced a weaker response in protoplasts than AvrSr35-01, consistent with the Sr35-virulent phenotype of ITA2018-1 (homozygous for *avrsr35-02*) (**Supplemental Figure 21A-C**). There are 40 single amino acid differences and 2 indels between AvrSr35-01 and -02, but only one of these lies in the interaction interface with Sr35^{41,42}, S349 (AvrSr35-01) to R352 (AvrSr35-02). Experimental data in *N. benthamiana* shows exchanging this single residue is sufficient to abolish recognition in AvrSr35-01 and restore recognition in AvrSr35-02 (**Supplemental Figure 21D-E**).”

References 41 and 42 were added to the literature cited.

L677-679

“Amino acid mutations were introduced by Dpn1-mediated site-directed mutagenesis using Phusion High-Fidelity DNA polymerase (NEB, Cat. #M0530L).”

L676: What *Agrobacterium* strain was used?

The *agrobacterium* strains used were GV3103-pMP90RK (pAM constructs) and GV3101-pMP90RK (pBIN19 constructs).

We have amended the main text as follows:

L676-L679:

“*Agrobacterium tumefaciens* cultures of strains GV3103-pMP90RK (pAM constructs) and GV3101-pMP90RK (pBIN19 constructs) were grown at 28°C overnight with shaking at 200 RPM in Luria-Bertani liquid medium with appropriate antibiotic selections.”

Reviewer #2 (Remarks to the Author):

The manuscript "Allelic variation of Avr genes explains virulence of wheat stem rust isolates associated with severe disease epidemics" provides chromosome-level and nuclear-phased genomes of two strains of *Puccinia graminis* f. sp. *tritici* (Pgt). These enabled the identification of two additional nuclear haplotypes, and the characterization of Avr variants. The work is very nicely written, provides very good resources and advances our understanding on Pgt virulence and evolution. I have very few and minor comments:

I would like to ask the authors to highlight more the novelty of the work. Several of the Avrs the authors characterized have already been identified. nuclear level sequencing was already obtained for Pgt. An additional Avr was cloned, but there are 6 that have already been cloned.

Thank you very much for reviewing our work, we appreciate your feedback. We have modified the abstract to better highlight the novelty and significance of the work, especially how the new genome references have allowed us to establish and populate an Avr gene atlas that represents the virulence genotypes of the major epidemic strains of this pathogen on a set of important resistance genes used in breeding. This is a significant development that will support breeding decisions to prioritise certain R genes and enable improved pathogen surveillance by incorporation of Avr genotyping into molecular diagnostic tools.

“Wheat stem rust is a disease of global importance caused by the fungal pathogen *Puccinia graminis* f. sp. *tritici* (*Pgt*). Since the emergence of *Pgt* strain Ug99 (race TTKSK) in 1998, additional *Pgt* races have emerged as food security threats. We generated chromosome level, nuclear-phased genome references for *Pgt* isolates ETH2013-1 and ITA2018-1, representing races TKTTF and TTRTF respectively, that have caused major epidemics in Africa and Europe. The nuclear haplotypes of ETH2013-1 and ITA2018-1 are unique and unrelated to those of Ug99 and Pgt21, indicating independent origins. The nuclear haplotypes showed extensive variation in sequence and copy number of six known *Avr* genes as well as for *AvrSr33*, which we identified through an effector gene library screen. We characterised the recognition properties of 22 novel *Avr* gene variants to establish an *Avr* gene atlas for *Pgt*. These results explained the race virulence phenotypes and epidemiology, such as the outbreak of TTRTF on durum cultivars containing *Sr13b* due to ITA2018-1 carrying a homozygous deletion of *AvrSr13*. This *Avr* gene atlas can inform wheat breeding, for instance, only *AvrSr22* is homozygous in all sequenced rust strains, suggesting *Sr22* may provide more durable resistance. This atlas will also enable development of sequence-based virulence diagnostic tools to enhance pathogen surveillance.”

How did the full genome sequencing of the isolates ITA2018-1 and helped the cloning of *AvrSr33*?

The effector library screen that enabled the identification of *AvrSr33* was constructed based on the genome annotation of the stem rust isolate *Pgt*21-0. The ETH2013-1 and ITA2018-1 isolates were then mined for variants of this gene for functional

characterisation to establish their genotypes with respect to avirulence/virulence for Sr33 and expand the Avr gene atlas.

Is there genetic exchange between the 2 nuclei of an isolate?

This is a very broad question that is not directly addressed by the data in this paper. The observations of nuclear exchange between isolates of Pgt and other cereal rusts show that these nuclei swaps occur without any exchange of genetic material between the two nuclear haplotypes of the parental isolates or between the nuclear haplotypes of the progeny. However, there is some evidence for movement of chromosomes between nuclei within an isolate – for instance Pgt21-0 contains both copies of chromosome 11 in one nucleus. As yet there is no direct evidence for crossing-over/recombination between chromosomes housed in different nuclei during asexual proliferation.